# SGD on Neural Networks Learns Functions of Increasing Complexity

**Preetum Nakkiran**
Harvard University

**Gal Kaplun**
Harvard University

**Dimitris Kalimeris**
Harvard University

**Tristan Yang**
Harvard University

**Benjamin L. Edelman**
Harvard University

**Fred Zhang**
Harvard University

**Boaz Barak**
Harvard University[*]

## Abstract

We perform an experimental study of the dynamics of Stochastic Gradient Descent (SGD) in learning deep neural networks for several real and synthetic classification tasks. We show that in the initial epochs, almost all of the performance improvement of the classifier obtained by SGD can be explained by a linear classifier. More generally, we give evidence for the hypothesis that, as iterations progress, SGD learns functions of increasing complexity. This hypothesis can be helpful in explaining why SGD-learned classifiers tend to generalize well even in the over-parameterized regime. We also show that the linear classifier learned in the initial stages is "retained" throughout the execution even if training is continued to the point of zero training error, and complement this with a theoretical result in a simplified model. Key to our work is a new measure of how well one classifier explains the performance of another, based on conditional mutual information.

## 1 Introduction

Neural networks have been extremely successful in modern machine learning, achieving the state-of-the-art in a wide range of domains, including image-recognition, speech-recognition, and game-playing [14, 18, 23, 37]. Practitioners often train deep neural networks with hundreds of layers and millions of parameters and manage to find networks with good out-of-sample performance. However, this practical prowess is accompanied by feeble theoretical understanding. In particular, we are far from understanding the *generalization performance* of neural networks—why can we train large, complex models on relatively few training examples and still expect them to generalize to unseen examples? It has been observed in the literature that the classical generalization bounds that guarantee small generalization gap (i.e., the gap between train and test error) in terms of VC dimension or Rademacher complexity do not yield meaningful guarantees in the context of real neural networks. More concretely, for most if not all real-world settings, there exist neural networks which fit the train set exactly, but have arbitrarily bad test error [41].

The existence of such "bad" empirical risk minimizers (ERMs) with large gaps between the train and test error means that the generalization performance of deep neural networks depends on the particular algorithm (and initialization) used in training, which is most often stochastic gradient descent (SGD). It has been conjectured that SGD provides some form of "implicit regularization" by outputting "low complexity" models, but it is safe to say that the precise notion of complexity and the mechanism by which this happens are not yet understood (see related works below).

---

[*]`preetum@cs.harvard.edu, galkaplun@g.harvard.edu, kalimeris@g.harvard.edu,`
`tristanyang@college.harvard.edu, bedelman@g.harvard.edu, hzhang@g.harvard.edu,`
`b@boazbarak.org`

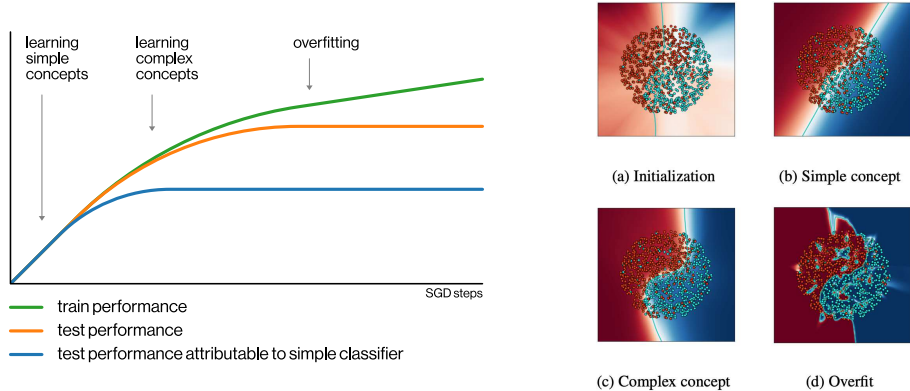

Figure 1: *Left:* An illustration of our hypothesis of how SGD dynamics progress. Initially, all progress in learning can be attributed to a "simple" classifier (in some precise sense to be later defined), then SGD continues in learning more complex but still meaningful classifiers. Finally, the classifier will interpolate the training data, while retaining correlation with simpler classifiers that allows it to generalize. *Right:* A plot of how the decision boundary evolves as a neural network is trained for a simple classification task. The data distribution is uniform in a 2-dimensional ball of radius 1, labeled by a sinusoidal curve with 10% label noise. It is evident that an almost linear decision boundary emerges in the first phases of training before more complex classifiers are learned. In the last stages, the network overfits to the label noise, while still retaining the concept.

In this paper, we provide evidence for this hypothesis and shed some light on how it comes about. Specifically, our thesis is that the *dynamics* of SGD play a crucial role and that SGD finds generalizing ERMs because:

(i) In the initial epochs of learning, SGD has a bias towards *simple classifiers* as opposed to complex ones; and

(ii) in later epochs, SGD is *relatively stable* and retains the information from the simple classifier it obtained in the initial epochs.

Figure 1 illustrates qualitatively the predictions of this thesis for the dynamics of SGD over time. In this work, we give experimental and theoretical evidence for both parts of this thesis. While several quantitative measures of complexity of neural networks have been proposed in the past, including the classic notions of VC dimension, Rademacher complexity and margin [2, 6, 20, 32, 22, 5], we do not propose such a measure here. Our focus is on the *qualitative* question of how much of SGD's early progress in learning can be explained by simple models. Our main findings are the following:

**Claim 1** (Informal). *In natural settings, the initial performance gains of SGD on a randomly initialized neural network can be attributed almost entirely to its learning a function correlated with a* linear classifier *of the data.*

**Claim 2** (Informal). *In natural settings, once SGD finds a simple classifier with good generalization, it is likely to retain it, in the sense that it will perform well on the fraction of the population classified by the simple classifier, even if training continues until it fits all training samples.*

We state these claims broadly, using "in natural settings" to refer to settings of network architecture, initialization, and data distributions that are used in practice. We emphasize that this holds for *vanilla* SGD with standard architecture and random initialization, without using any regularization, dropout, early stopping or other explicit methods of biasing towards simplicity.

Some indications for variants of Claim 2 have been observed in practice, but we provide further experimental evidence and also show (Theorem 1) a simple setting where it *provably* holds. Our main novelty is Claim 1, which is established via several experiments described in Sections 3 and 4. We emphasize that our claims *do not* imply that during early stages of training the decision boundary is linear, but rather that there often exists a linear classifier whose correct predictions highly agree with the network's correct predictions. The decision boundary itself may be very complex.[2]

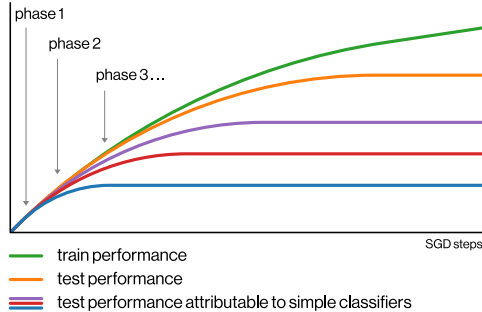

phase 1
phase 2
phase 3...

SGD steps

— train performance
— test performance
≡ test performance attributable to simple classifiers

Figure 2: *Beyond linear classifiers.* The two phases of SGD learning in Figure 1 can be broken into several sub-phases. Phase $i$ involves learning classifiers of lower "complexity" than phase $i+1$. The precise notion of complexity may be algorithm, initialization and architecture-dependent. In practice, we expect that the phases will not be completely disjoint and some learning of classifiers of differing complexity will co-occur at the same time.

The other core contribution of this paper is a novel formulation of a *mutual-information based measure* to quantify how much of the prediction success of the neural network produced by SGD can be attributed to a simple classifier. We believe this measure is of independent interest.

**Remark 1** (Beyond linear classifiers). *While our main findings relate to linear classifiers, our methodology extends beyond this. We conjecture that generally, the dynamics of SGD are such that it initially learns simpler components of its final classifier, and retains these as it continues to learn more and more complex parts (see Figure 2). We provide evidence for this conjecture in Section 4.*

**Remark 2** (Beyond binary classification). *This paper is focused on binary classification tasks but our mutual-information based definitions and methodology can be extended to multi-class classification. Preliminary results suggest that our results continue to hold.*

**Related Work.**   There is a substantial body of work that attempts to understand the generalization of (deep) neural networks, tackling the problem from different perspectives. Previous works by Hardt et. al. (2016) and Kuzborskij & Lampert (2017) [17, 24] argue that generalization is due to stability. Neyshabur et. al. (2015); Keskar et. al. (2016); Bartlett et. al. (2016) consider margin-based approaches [32, 22, 5], while Dziugaite & Roy (2017); Neyshabur et. al. (2017); Neyshabur et. al. (2018); Golowich et. al. (2018); Pérez et. al. (2019); Zhou et. al. (2019) focus on PAC-Bayes analysis and norm-based bounds [10, 31, 30, 12, 34, 42]. Arora et. al. (2018) [3] propose a compression-based approach.

The implicit bias of (stochastic) gradient descent was also studied in various contexts, including linear classification, matrix factorization and neural networks. This includes the works of Brutzkus et. al. (2017); Gunasekar et. al. (2017); Soudry et. al. (2018); Gunasekar et. al. (2018); Li et. al. (2018); Wu et. al. (2019) and Ji & Telgarsky (2019) [9, 16, 38, 15, 26, 39, 21]. There are also recent works proving generalization of overparameterized networks, by analyzing the specific behavior of SGD from random initialization [1, 8, 25]. These results are so far restricted to simplified settings.

Several prior works propose measures of the complexity of neural networks, and claim that training involves learning simple patterns [4, 40, 35, 33]. However, our formalization has many advantages over prior formalizations. A key difference is that our measures are intrinsic to the classification function and data-distribution (and do not depend on the representation of the classifier, or its behavior outside the data distribution). Moreover, our measures address the extent by which one classifier "explains" the performance of another. Finally, our metrics are tractable to estimate in high dimensions, and are experimentally demonstrated for real-world distributions.

Most similar to our work is a concurrent work by Mangalam and Prabhu that also experimentally demonstrates that neural networks trained with SGD first learn to be able to classify examples that are learnable by simpler models. Their focus is on the complexity of the *examples*, not the learned functions, and their metrics are different.

The concept of mutual information has also been used in the study of neural networks, though in different ways than ours. For example, Schwartz-Ziv and Tishby (2017) [36] use it to argue that a network compresses information, saving only the most meaningful representation of the input.

**Paper Organization.**   We begin by defining our mutual-information based formalization of Claims 1 and 2 in Section 2. In Section 3, we establish the main result of the paper—that for many synthetic and real data sets, the performance of neural networks in the early phase of training is well explained by a linear classifier. In Section 4, we investigate extensions to non-linear classifiers (see also

Remark 1). We make the case that as training proceeds, SGD moves beyond this "linear learning" regime, and learns concepts of increasing complexity. In Section 5 we focus on the overfitting regime. We provide a simple theoretical setting where, provably, if we start from a "simple" generalizable solution, then overfitting to the train set will not hurt generalization. Moreover, the overfit classifier retains the information from the initial classifier. Finally, in Section 6 we discuss future directions.

## 2 *Performance Correlation* via Mutual Information

In this section, we present our measures for the contribution of a "simple classifier" to the performance of a "more complex" one. This allows us to state what it means for the performance of a neural network to be "almost entirely explained by a linear classifier", formalizing Claims 1 and 2.

### 2.1 Notation and Preliminaries

Key to our formalism are the quantities of mutual information and conditional mutual information. Recall that for three random variables $X, Y, Z$, the *mutual information* between $X$ and $Y$ is defined as $I(X; Y) = H(Y) - H(Y|X)$ and the *conditional mutual information* between $X$ and $Y$ conditioned on $Z$ is defined as $I(X; Y|Z) = H(Y|Z) - H(Y|X, Z)$, where $H$ is the (conditional) entropy.

We consider a joint distribution $(X, Y)$ on data and labels $(\mathcal{X}, \mathcal{Y}) \subseteq \mathbb{R}^d \times \{0, 1\}$. For a classifier $f : \mathcal{X} \to \mathcal{Y}$ we use the capital letter $F$ to denote the random variable $f(X)$. While the standard measure of prediction success is the accuracy $\mathbb{P}[F = Y]$, we use the mutual information $I(F; Y)$ instead. This makes no qualitative difference since the two are monotonically related (see Figure 3). In all plots, we plot the corresponding accuracy axis on the right for ease of use. We use $(X_S, Y_S)$ for the empirical distribution over the training set, and use $I(F; Y_S)$ (with a slight abuse of notation) for the mutual information between $f(X_S)$ and $Y_S$, a proxy for $f$'s success on the training set.

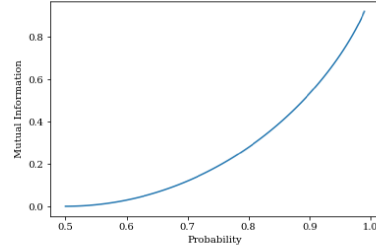

Figure 3: $I(F; Y)$ as a function of $\mathbb{P}[F = Y]$ for unbiased binary $F, Y$ s.t. $\mathbb{P}[F = Y] \geq 1/2$.

### 2.2 Performance correlation

If $f$ and $\ell$ are classifiers, and as above $F$ and $L$ are the corresponding random variables, then the chain rule for mutual information implies that[3]

$$I(F; Y) = I(F; Y|L) + I(L; Y) - I(L; Y|F).$$

We interpret the quantity $I(F; Y|L)$ as capturing the part of the success of $f$ on predicting $Y$ that cannot be explained by the classifier $\ell$. For example, $I(F; Y|L) = 0$ if and only if the prediction $f(X)$ is conditionally independent of the label $Y$, when given $\ell(X)$. In general, $I(F; Y|L)$ is the amount by which knowing $f(X)$ helps in predicting $Y$, given that we already know $\ell(X)$. Based on this interpretation, we introduce the following definition:

**Definition 1.** *For random variables $F, L, Y$ we define the* performance correlation *of $F$ and $L$ as*

$$\mu_Y(F; L) := I(F; Y) - I(F; Y|L) = I(L; Y) - I(L; Y|F) = I(F; L) - I(F; L|Y).$$

The performance correlation is always upper bounded by the minimum of $I(L; Y)$, $I(F; Y)$, and $I(F; L)$.[4] If $\mu_Y(F; L) = I(F; Y)$ then $I(F; Y|L) = 0$ which means that $f$ does not help in predicting $Y$, if we already know $\ell$. Hence, when $\ell$ is a "simpler" model than $f$, we consider $\mu_Y(F; L)$ as denoting the part of $F$'s performance that can be attributed to $\ell$.[5]

The reason why we use $\mu_Y(F; L)$ instead of simply $I(F; L)$ is the following. While it is true that $\mu_Y(F; L) \leq I(F; L)$, $\mu_Y$ captures the degree to which the information learned by $F$ *about* $Y$ is explained by $L$. whereas $I(F; L)$ only captures the correlation of $F$ and $L$, regardless of whether this correlation is useful for predicting $Y$ or not. For example, consider a scenario where $F(x) = L(x) \cdot Bernoulli(p)$. That is, $F$ is a linear classifier $L$ with noisy outputs. Here, $I(F; L) \ll 1$, due to the noise in $F$. Hence we might infer that $F$ does not agree with $L$. However, $\mu_Y(F; L) = I(F; Y)$, i.e. our metric recovers the fact that all the performance of $F$ in predicting $Y$ is explained by $L$.

Throughout this paper, we denote by $f_t$ the classifier SGD outputs on a randomly-initialized neural network after $t$ gradient steps, and denote by $F_t$ the corresponding random variable $f_t(X)$. We now formalize Claim 1 and Claim 2:

**Claim 1** ("Linear Learning", Restated). *In natural settings, there is a linear classifier $\ell$ and a step number $T_0$ such that for all $t \leq T_0$, $\mu_Y(F_t; L) \approx I(F_t; Y)$. That is, almost all of $f_t$'s performance is explained by $\ell$. Furthermore at $T_0$, $I(F_{T_0}; Y) \approx I(L; Y)$. That is, this initial phase lasts until $f_t$ approximately matches the performance of $\ell$.*

**Claim 2** (Restated) . *In natural settings, for $t > T_0$, $\mu_Y(F_t; L)$ plateaus at value $\approx I(L; Y)$ and does not shrink significantly even if training continues until SGD fits all the training set.*

# 3   SGD Learns a Linear Model First

In this section, we provide experimental evidence for Claim 1—the first phase of SGD is dominated by "linear learning"—and Claim 2—at later stages SGD retains information from early phases. We demonstrate these claims by evaluating our information-theoretic measures empirically on real and simulated classification tasks.

**Experimental Setup.**   We provide a brief description of our experimental setup here; a full description is provided in Appendix B. We consider the following binary classification tasks [6]:

   (i) Binary MNIST: predict whether the image represents a number from $0$ to $4$ or from $5$ to $9$.

   (ii) CIFAR-10 Animals vs Objects: predict whether the image represents an animal or an object.

   (iii) CIFAR-10 First 5 vs Last 5: predict whether the image is in classes $\{0 \ldots 4\}$ or $\{5 \ldots 9\}$.

   (iv) High-dimensional sinusoid: predict $y := \text{sign}(\langle \boldsymbol{w}, \boldsymbol{x} \rangle + \sin\langle \boldsymbol{w}', \boldsymbol{x} \rangle)$ for standard Gaussian $\boldsymbol{x} \in \mathbb{R}^{100}$, and $\boldsymbol{w} \perp \boldsymbol{w}'$.

We train neural networks with standard architectures: CNNs for image-recognition tasks and Multi-layer Perceptrons (MLPs) for the other tasks. We use standard uniform Xavier initialization [11] and we train with binary cross-entropy loss. In all experiments, we use *vanilla SGD* without regularization (e.g., dropout, weight decay) for simplicity and consistency. (Preliminary experiments suggest our results are robust with respect to these choices). We use a relatively small step-size for SGD, in order to more closely examine the early phase of training.

In all of our experiments, we compare the classifier $f_t$ output by SGD to a linear classifier $\ell$. If the population distribution has a unique optimal linear classifier $\ell^*$ then we can use $\ell = \ell^*$. This is the case in tasks (i),(ii),(iv). If there are different linear classifiers that perform equally well (task (iii)), then the classifier learned in the first stage could depend on the initialization. In this case, we pick $\ell$ by searching for the linear classifier that best fits $f_{T_0}$, where $T_0$ is the step in which $I(F_t; Y)$ reaches the best linear performance $\max_{L'} I(L'; Y)$. In either case, it is a highly non-trivial fact that there is *any* linear classifier that accounts for the bulk of the performance of the SGD-produced classifier $f_t$.

**Results and Discussion.**   The results of our experiments are presented in Figure 4. We observe the following similar behaviors across several architectures and datasets:

Define the *first phase* of training as all steps $t \leq T_0$, where $T_0$ is the first SGD step such that the network's performance $I(F_t; Y)$ reaches the linear model's performance $I(L; Y)$. Now:

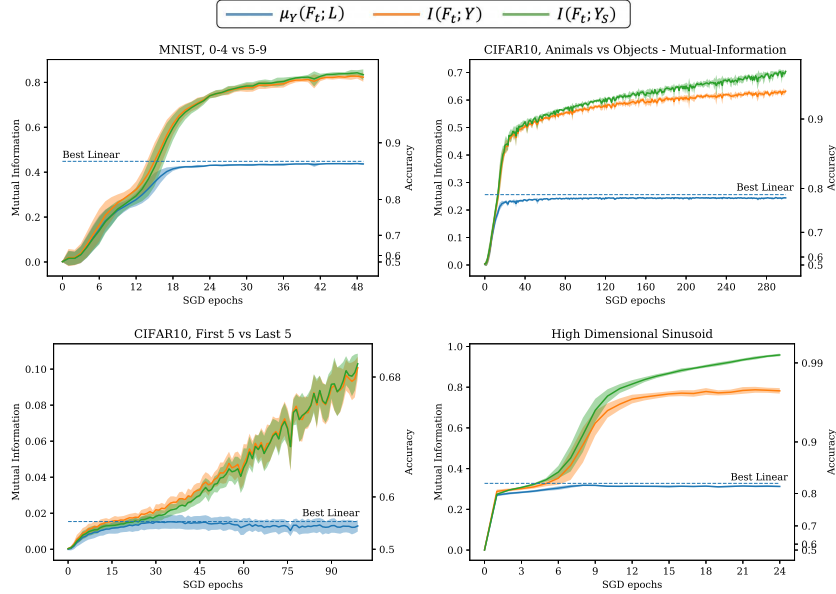

Figure 4: SGD dynamics for various classification tasks. In each figure, we plot both the value of the mutual information and the corresponding accuracy. Observe that in the initial phases the bulk of the increase in performance is attributed to the linear classifier, since $\mu_Y(F; L) \approx I(F_t; Y)$.

1. During the first phase of training, $\mu_Y(F_t; L)$ is close to $I(F_t; Y)$ thus, most of the performance of $F_t$ can be attributed to $\ell$. In fact, we can often pick $\ell$ such that $I(L; Y)$ is close to $\max_{L'} I(L'; Y)$, the performance of the best linear classifier for the distribution. In this case, the fact that $\mu_Y(F_{T_0}; L) \approx I(F_{T_0}; Y) \approx \max_{L'} I(L'; Y)$ means that SGD not only starts learning a linear model, but remains in the "linear learning" regime until it has learnt almost the best linear classifier. Beyond this point, the model $F_t$ cannot increase in performance without learning more non-linear aspects of $Y$.

2. In the following epochs, for $t > T_0$, $\mu_Y(F_t; L)$ plateaus around $I(L; Y)$. This means that $F_t$ retains its correlation with $L$, which keeps explaining as much of $F_t$'s generalization performance as possible.

Observation (1) provides strong support for Claim 1. Since neural networks are a richer class than linear classifiers, a priori one might expect that throughout the learning process, some of the growth in the mutual information between the label $Y$ and the classifier's output $F_t$ will be attributable to the linear classifier, and some of this growth will be attributable to a more complex classifier. However, what we observe is a relatively clean (though not perfect) separation of the learning process while in the initial phase, all of the mutual information between $F_t$ and $Y$ disappears if we condition on $L$.

To understand this result's significance, it is useful to contrast it with a "null model" where we replace the linear classifier $\ell$ by a random classifier $\tilde{\ell}$ having the same mutual information with $Y$ as $\ell$.[7] Now, consider the ratio $\mu_Y(F_{T_0}; \tilde{L})/I(F_{T_0}; Y)$ at the end of the first phase. It can be shown that this ratio is small, meaning that the performance of $F_t$ is not well explained by $\tilde{L}$. However, in our case with a linear classifier, this ratio is much closer to 1 at the end of the first phase. For example, for CIFAR (iii), the linear model $L$ has $\mu_Y(F_{T_0}; L)/I(F_{T_0}; Y) = 0.80$ while the corresponding null model $\tilde{L}$ has ratio $\mu_Y(F_{T_0}; \tilde{L})/I(F_{T_0}; Y) = 0.31$. This illustrates that the early stage of learning is biased specifically towards linear functions, and not towards arbitrary functions with non-trivial accuracy. Similar metrics for all datasets are reported in Table 2 in the Appendix.

Observation (2) can be seen as offering support to Claim 2. If SGD "forgets" the linear model as it continues to fit the training examples, then we would expect the value of $\mu_Y(F_t; L)$ to *shrink* with time. However, this does not occur. Since the linear classifier itself would generalize, this

explains at least part of the generalization performance of $F_t$. To fully explain the generalization performance, we would need to extend this theory to models more complex than linear; some preliminary investigations are given in Section 4.

Table 1 summarizes the qualitative behavior of several information theoretic quantities we observe across different datasets and architectures. We stress that these phenomena would not occur for an arbitrary learning algorithm that increases model test accuracy. Rather, it is SGD (with a random, or at least "non-pathological" initialization, see Section 5) that produces such behavior. The initialization is important since in Figure 8 in the appendix we show that one can construct adversarial initializations for which this inductive bias of SGD breaks. Concurrent work by Liu et al. [27] also finds an initialization for SGD that leads to poor generalization, using a slightly different technique.

|  | Train acc | Test acc | $I(F_t;Y)$ | $\mu_Y(F_t;L)$ | $I(F_t;Y\mid L)$ | $I(L;Y\mid F_t)$ |
|---|---|---|---|---|---|---|
| First phase | $\uparrow$ | $\uparrow$ | $\uparrow$ | $\approx I(F_t;Y)$<br>increase in acc of $F_t$ explained by $L$ | $\approx 0$ | $\downarrow$<br>$F_t$ starts correlating with $L$ |
| Middle phase | $\uparrow$ | $\uparrow$ | $\uparrow$ | $-$<br>plateaus near $I(L;Y)$ | $\uparrow$<br>$F_t$ becomes more expressive than $L$ | $\approx 0$<br>$F_t$ doesn't forget $L$ |
| Overfitting | $\uparrow$<br>overfit to train set | $-$<br>overfitting doesn't hurt or improve test | $-$ | $-$ | $-$ | $\approx 0$<br>$F_t$ still doesn't forget $L$ |

Table 1: Qualitative behavior of the quantities of interest in our experiments. We denote with $\uparrow$, $\downarrow$ and $-$ increasing, decreasing and constant values respectively.

## 4  Beyond Linear: SGD Learns Functions of Increasing Complexity

In this section we investigate Remark 1—that SGD learns functions of increasing complexity— through the lens of the mutual information framework, and provide experimental evidence supporting the natural extension of the results from Section 3 to models more complex than linear.

**Conjecture 1** (Beyond linear classifiers: Remark 1 restated). *There exist increasingly complex functions* $(g_1, g_2, ...)$ *under some measure of complexity, and a monotonically increasing sequence* $(T_1, T_2, ...)$ *such that* $\mu_Y(F_t;G_i) \approx I(F_t;Y)$ *for* $t \leq T_i$ *and* $\mu_Y(F_t;G_i) \approx I(G_i;Y)$ *for* $t > T_i$. [8]

It is problematic to show Conjecture 1 in full generality, as the correct measure of complexity is unclear; it may depend on the distribution, architecture, and even initialization. Nevertheless, we are able to support it in the image-classification setting, parameterizing complexity using the number of convolutional layers.

**Experimental Setup.**  In order to explore the behavior of more complex classifiers we consider the CIFAR "First 5 vs. Last 5" task introduced in Section 3, for which there is no high-accuracy linear classifier. We observed that the performance of various architectures on this task was similar to their performance on the full 10-way CIFAR classification task, which supports the relevance of this example to standard use-cases.[9]

As our model $f$, we train an 18-layer pre-activation ResNet described in [19] which achieves over 90% accuracy on this task. For the simple models $g_i$, we use convolutional neural networks corresponding to the 2nd, 4th, and 6th shallowest layers of the network for $f$. Similarly to Section 3, the models $g_i$ are trained on the images labeled by $f_\infty$ (that is the model at the end of training). For more details refer to Appendix B: "Finding the Conditional Models".

**Results and Discussion.**  Our results are illustrated in Figure 5. We can see a *separation in phases for learning*, where all curves $\mu_Y(F_t;G_i)$ are initially close to $I(F_t;Y)$, before each successively plateaus as training progresses. Moreover, note that $I(G_i;Y)$ remains flat in the overfitting regime for all three $i$, demonstrating that SGD does not "forget" the simpler functions as stated in Claim 2.

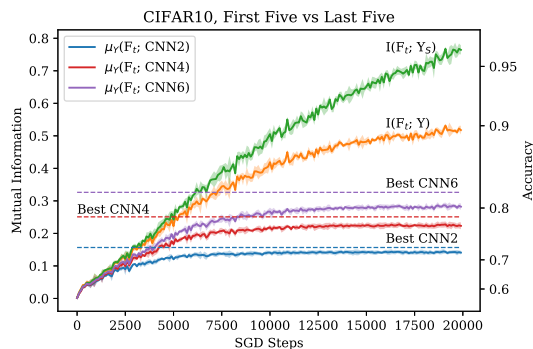

Figure 5: Distinguishing between the first vs. the last 5 classes of CIFAR10. CNN$k$ denotes a convolutional neural network of $k$ layers. We clearly see a *separation in phases of learning*, where all curves $\mu_Y(F_t; G_i)$ are initially close to $I(F_t, Y)$, before each successively plateaus as training progresses. The plot matches the conjectured behavior illustrated in Figure 2.

Interestingly, the 4 and 6-layer CNNs exhibit less clear phase separation than the 2-layer CNN and linear model of Section 3. We attribute this to two possibilities—firstly, training $g_i$ on $f_\infty$ for larger models likely may not recover the *best* possible simple classifier that explains $f_t$ [10]; secondly, the number of layers may not be a perfect approximation to the notion of simplicity. However, we can again verify our qualitative results by comparing to a random "null model" classifier $\tilde{g}_i$ with the same accuracy as $g_i$. For the 6-layer CNN, $\mu(F_{T_0}; G_i)/I(F_{T_0}; Y) = 0.72$, while $\mu(F_{T_0}; \widetilde{G}_i)/I(F_{T_0}; Y) = 0.40$, with $T_0$ estimated as before (see Table 3 in Appendix B for the 2 and 4-layer numbers). Thus, $g_i$ explains the behavior of $f_t$ significantly more than an arbitrary classifier of equivalent accuracy.

## 5   Overfitting Does Not Hurt Generalization

In the previous sections we investigated the early and middle phases of SGD training. In this section, we focus on the last phase, i.e. the overfitting regime. In practice, we often observe that in late phases of training, train error goes to 0, while test error stabilizes, despite the fact that bad ERMs exist. The previous sections suggest that this phenomenon is an inherent property of SGD in the overparameterized setting, where training starts from a "simpler" model at the beginning of the overfitting regime and does not forget it even as it learns more "complex" models and fits the noise.

In what follows, we demonstrate this intuition formally in an illustrative simplified setting where, *provably*, a heavily overparameterized (linear) model trained with SGD fits the training set exactly, and yet its population accuracy is optimal for a class of "simple" initializations.[11]

**The Model.** We confine ourselves to the linear classification setting. To formalize notions of "simple" we consider a data distribution that explicitly decomposes into a component explainable by a *sparse* classifier, and a remaining orthogonal noisy component on which it is possible to overfit. Specifically, we define the data distribution $\mathcal{D}$ as follows:

$$y \overset{u.a.r.}{\sim} \{-1, +1\}, \qquad \boldsymbol{x} = \eta \cdot y \cdot \boldsymbol{e}_1 + \boldsymbol{e}_k, \qquad k \overset{u.a.r.}{\sim} \{2, \dots, d\},$$
$$\eta \sim \text{Bernoulli}(p) \text{ over } \{\pm 1\}.$$

Here $\boldsymbol{e}_i$ refers to the $i^{th}$ vector of the standard basis of $\mathbb{R}^d$, while $p \leq 1/2$ is a noise parameter. For $1 - p$ fraction of the points the first coordinate corresponds to the label, but a $p$ fraction of the points are "noisy", i.e., their label is the opposite of their first coordinate. Notice that the classes are essentially linearly separable up to error $p$.

We deal with the heavily overparameterized regime, i.e., when we are presented with only $n = o(\sqrt{d})$ samples. We analyze the learning of a linear classifier $\boldsymbol{w} \in \mathbb{R}^d$ by minimizing the empirical square loss $\mathcal{L}(\boldsymbol{w}) = \frac{1}{n} \sum_{i=1}^{n} (1 - y_n \langle \boldsymbol{w}, \boldsymbol{x}_i \rangle)^2$ using SGD. Key to our setting is the existence of *poor* ERMs—classifiers that have $\leq 50\%$ population accuracy but achieve 100% training accuracy by taking advantage of the $\boldsymbol{e}_k$ components of the sample points, which are noise, not signal. We show

however, that as long as we begin not too far from the "simplest" classifier $\boldsymbol{w}^* = \boldsymbol{e}_1$, the ERM found by SGD generalizes well. This holds empirically even for more complex models (Fig 8 in App C).

**Theorem 1.** *Consider training a linear classifier via minimizing the empirical square loss using SGD. Let $\varepsilon > 0$ be a small constant and let the initial vector $\boldsymbol{w}_0$ satisfy $\boldsymbol{w}_0(1) \geq -n^{0.99}$, and $|\boldsymbol{w}_0(i)| \leq 1 - 2p - \varepsilon$ for all $i > 1$. Then, with high probability, sample accuracy approaches 1 and population accuracy approaches $1 - p$ as the number of gradient steps goes to infinity.*

*Proof sketch.* The displacement of the weight vector from initialization will always lie in the span of the sample vectors which, because the samples are sparse, is in expectation almost orthogonal to the population. Moreover, as long as the initialization is bounded sufficiently, the first coordinate of the learned vector will approach a constant. The full proof is deferred to Appendix A. □

Theorem 1 implies in particular that if we initialize at a good bounded model (such as $\boldsymbol{w}^*$), a version of Claim 2 provably applies to this setting: if $F_t$ corresponds to the model at SGD step $t$ and $\ell$ corresponds to $\boldsymbol{w}^*$, then $\mu_Y(F_t; L)$ will barely decrease in the long term.

## 6   Discussion and Future Work

Our findings yield new insight into the inductive bias of SGD on deep neural networks. In particular, it appears that SGD increases the complexity of the learned classifier as training progresses, starting by learning an essentially linear classifier.

There are several natural questions that arise from our work. First, *why* does this "linear learning" occur? We pose this problem of understanding why Claims 1 and 2 are true as an important direction for future work. Second, what is the correct measure of complexity which SGD increases over time? That is, we would like the correct formalization of Conjecture 1—ideally with a measure of complexity that implies generalization. We view our work as an initial step in a framework towards understanding why neural networks generalize, and we believe that theoretically establishing our claims would be significant progress in this direction.

**Acknowledgements.** We thank all of the participants of the Harvard ML Theory Reading Group for many useful discussions and presentations that motivated this work. We especially thank: Noah Golowich, Yamini Bansal, Thibaut Horel, Jarosław Błasiok, Alexander Rakhlin, and Madhu Sudan.

This work was supported by NSF awards CCF 1565264, CNS 1618026, CCF 1565641, CCF 1715187, NSF GRFP Grant No. DGE1144152, a Simons Investigator Fellowship, and Investigator Award.

## Footnotes

[2]Figure 6 in Appendix C provides a simple illustration of this phenomenon.

[3]Specifically, the equation can be derived by using the chain rule $I(A, B; C) = I(B; C|A) + I(A; C)$ to express $I(F, L; Y)$ as both $I(F; Y|L) + I(L; Y)$ and $I(L; Y|F) + I(F; Y)$.

[4]The quantity $\mu_Y(F, L)$ can also be thought as a multivariate generalization of mutual information [28, 7].

[5]This interpretation is slightly complicated by the fact that, like correlation, $\mu_Y(F; L)$ can sometimes be negative. However, this quantity is always non-negative under various weak assumptions which hold in practice, e.g. when both $F$ and $L$ have significant test accuracy, or when $H(Y|F, L) \geq \min\{H(Y|F), H(Y|L)\}$.

[6]We focus on binary classification because: (1) there is a natural choice for the "simplest" model class (i.e., linear models), and (2) our mutual-information based metrics can be more accurately estimated from samples. We have preliminary work extending our results to the multi-class setting.

[7]That is, $\tilde{\ell}(X) = Y$ with probability $p$ and random otherwise, where $p$ is set to ensure $I(\tilde{L}; Y) = I(L; Y)$.

[8]Note that implicit in our conjecture is that each $G_i$ is itself explained by $G_{<i}$, so we should not have to condition on all previous $G_i$'s; i.e. $\mu_Y(F_t;(G_{1:i})) \approx \mu_Y(F_t;G_i)$.

[9]Potentially since we need to distinguish between visually similar classes, e.g. automobile/truck or cat/dog.

[10]In the extreme case, $g_i$ has same architecture as $f$. We cannot recover $f$ exactly by training on its outputs.

[11]A similar setting is analyzed in the concurrent work of Nagarajan and Kolter [29] to show the limitations of uniform convergence bounds for explaining generalization of deep learning.

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
