[Supplementary Material · appendix.pdf]

## A    Proof of Theorem 1

For the simplicity of our argument, we work with the following assumptions on the training set.

**Assumption 1.** *Label noise: Exactly $p$ fraction of the sample points have their first coordinate flipped.*

**Assumption 2.** *Orthogonality: the non-zero coordinates are distinct for all $n$ data points (except for the first coordinate)*

Notice that by the fact that $n = o(\sqrt{d})$ and a simple union bound, Assumption 2 holds with high probability. For each $i \in [d]$, we let $j(i)$ denote the index $j$ that satisfies $\boldsymbol{x}_j(i) = 1$, if it exists. To simplify the notation, we assume that all labels $y_i$ are 1; this is without loss of generality, since one can always replace $\boldsymbol{x}_i$ with $y_i\boldsymbol{x}_i$.

In order to prove Theorem 1, we will precisely characterize the limiting behavior of SGD in this setting. We remark that as the optimization objective is strongly convex, SGD and GD with appropriate choices of step size is guaranteed to converge to the global minimum.

**Lemma 1** (Convergence of gradient descent). *Assume the setting above. Let $\boldsymbol{X} \in \mathbb{R}^{n \times d}$ be the data matrix. Initialized at point $\boldsymbol{w}_0$, (stochastic) gradient descent converges to*

$$\boldsymbol{w}' = \boldsymbol{X}^T(\boldsymbol{X}\boldsymbol{X}^T)^{-1}(\boldsymbol{1} - \boldsymbol{X}\boldsymbol{w}_0) + \boldsymbol{w}_0, \tag{1}$$

*as the number of steps goes to infinity. Moreover, we have*

$$\boldsymbol{w}'(1) = \left(\frac{n}{n+1}\right)\eta - \frac{\boldsymbol{s}^T\boldsymbol{X}\boldsymbol{w}_0}{n+1} + \boldsymbol{w}_0(1), \tag{2}$$

*where $\boldsymbol{s}$ is the first column of $\boldsymbol{X}$ and $\eta = 1 - 2p$, and for $i \neq 1$*

$$\boldsymbol{w}'(i) = \begin{cases} \boldsymbol{w}_0(i) & \text{if } \boldsymbol{x}_j(i) = 0 \text{ for all } \boldsymbol{x}_j, \\ \beta\left(1 + \boldsymbol{s}_{j(i)}\left(\frac{-n\eta + \boldsymbol{s}^T\boldsymbol{X}\boldsymbol{w}_0}{n+1}\right)\eta - \boldsymbol{x}_{j(i)}^T\boldsymbol{w}_0\right) + \boldsymbol{w}_0(i) & \text{if } \boldsymbol{x}_j(i) = \beta \in \{\pm 1\} \text{ for some } \boldsymbol{x}_j \end{cases} \tag{3}$$

*Proof.* We focus on the gradient descent case; the proof for SGD is analogous. Consider the empirical loss $\mathcal{L}(\boldsymbol{w})$. By the definition of gradient descent, the iterations always stay in the affine space $V = \boldsymbol{w}_0 + \text{RowSpan}(X)$. Gradient descent solves the linear least squares problem $\min_{\boldsymbol{w} \in V} \mathcal{L}(\boldsymbol{w})$. We claim that $w'$ is indeed the optimal solution to this program, and thus gradient descent converges to it.

First, one can check $\boldsymbol{X}\boldsymbol{X}^T$ is non-singular under Assumption 2, since $\boldsymbol{X}\boldsymbol{X}^T = \boldsymbol{I} + \boldsymbol{s}\boldsymbol{s}^T$ and $\boldsymbol{s}^T\boldsymbol{s} \neq -1$. Moreover, $\boldsymbol{w}' \in V$ since $\boldsymbol{X}^T(\boldsymbol{X}\boldsymbol{X}^T)^{-1}$ is the orthogonal projector onto the row space of $\boldsymbol{X}$. Finally, we check that $\boldsymbol{w}'$ achieves zero empirical loss

$$\boldsymbol{X}\boldsymbol{w}' = \boldsymbol{X}\boldsymbol{X}^T(\boldsymbol{X}\boldsymbol{X}^T)^{-1}(\boldsymbol{1} - \boldsymbol{X}\boldsymbol{w}_0) + \boldsymbol{X}\boldsymbol{w}_0 = \boldsymbol{1}. \tag{4}$$

For the first coordinate of $\boldsymbol{w}'$, by the Sherman-Morrison formula [12, p. 51],

$$(\boldsymbol{X}\boldsymbol{X}^T)^{-1} = (\boldsymbol{I} + \boldsymbol{s}\boldsymbol{s}^T)^{-1} = \boldsymbol{I} - \frac{\boldsymbol{s}\boldsymbol{s}^T}{1 + \boldsymbol{s}^T\boldsymbol{s}} = \boldsymbol{I} - \frac{\boldsymbol{s}\boldsymbol{s}^T}{n+1}. \tag{5}$$

Substituting this into (1) and simplifying yields claims (2) and (3). □

We now recall Theorem 1:

**Theorem 1.** *Consider training a linear classifier via minimizing the empirical square loss using SGD. Let $\varepsilon > 0$ be a small constant and let the initial vector $\boldsymbol{w}_0$ satisfy $\boldsymbol{w}_0(1) \geq -n^{0.99}$, and $|\boldsymbol{w}_0(i)| \leq 1 - 2p - \varepsilon$ for all $i > 1$. Then, with high probability, sample accuracy approaches 1 and population accuracy approaches $1 - p$ as the number of gradient steps goes to infinity.*

*Proof.* Let $k(j)$ denote the non-zero coordinate of $\boldsymbol{x}_j$ (besides the first coordinate). First we note that

$$\boldsymbol{s}^T\boldsymbol{X}\boldsymbol{w}_0 = n\boldsymbol{w}_0(1) + \sum_{j=1}^{n} \boldsymbol{s}_j\boldsymbol{x}_j(k(j))\boldsymbol{w}_0(k(j))$$

Because each $\boldsymbol{x}_j(k(j))$ is a Bernoulli$(p)$ random variable and every $\boldsymbol{w}_0(k(j)) \leq 1$, we can apply Chebyshev's inequality to further deduce that with high probability, $\boldsymbol{s}^T \boldsymbol{X} \boldsymbol{w}_0 = n\boldsymbol{w}_0(1) + O(\sqrt{n})$. Substituting this into (2) of Lemma 1, letting $\boldsymbol{w}'$ be the weight vector SGD converges to, we obtain

$$\boldsymbol{w}'(1) = \left(\frac{n}{n+1}\right)\eta - \frac{n\boldsymbol{w}_0(1) + O(\sqrt{n})}{n+1} + \boldsymbol{w}_0(1) = \eta - o(1)$$

By (3) of Lemma 1, for every coordinate $i$ such that $\boldsymbol{x}_j(i) = 0$ for all $j$, we have $\boldsymbol{w}'(i) = \boldsymbol{w}_0(i)$. Consider a point $\boldsymbol{x}$ drawn from the population, and let $k$ be the index of the non-zero coordinate of $\boldsymbol{x}$. With high probability, $k \neq k(j)$ for all $j \in [n]$. With probability $1 - p$, $\boldsymbol{x}(1) = 1$, and in this case we obtain

$$\langle \boldsymbol{w}', \boldsymbol{x} \rangle = \boldsymbol{w}'(1)\boldsymbol{x}(1) + \boldsymbol{w}_0(k)\boldsymbol{x}(k) \geq \eta - o(1) - (\eta - \varepsilon) = \varepsilon - o(1)$$

For sufficiently large $n$ (corresponding to sufficiently large $d$) this quantity is always positive, so with probability approaching $1 - p$ the model correctly classifies $\boldsymbol{x}$. $\qquad\square$

# B    Experimental Setup and Results for Sections 3 and 4

**Dataset description.**    We used the following four datasets in our experiments.

   (i) Binary MNIST: predict whether the image represents a number from 0 to 4 or from 5 to 9. It admits a linear classifier with accuracy $\approx 87\%$,

  (ii) CIFAR-10 Animals vs Objects: predict whether the image represents an animal or an object. In order not to enforce bias towards any of the classes we included all the 4 object classes (airplane, automobile, ship, truck) and only 4 out of 6 of the animal ones (bird, cat, dog, horse). Hence the number of positive and negative samples are the same. CIFAR-10 Animals vs Objects admits a linear classifier with accuracy $\approx 75\%$,

 (iii) CIFAR-10 First 5 vs Last 5: predict whether an image belongs to any of the first 5 classes of CIFAR10 (airplane, automobile, bird, cat, deer) or the last 5 classes (dog, frog, horse, ship, truck). CIFAR-10 First 5 vs Last 5 does not admit a linear classifier with satisfying accuracy. The best linear classifier achieves accuracy of $\approx 58\%$,

  (iv) High-dimensional Sinusoid: predict $y := \text{sign}(\langle w, x \rangle + \sin\langle w', x \rangle)$ for standard Gaussian $x \in \mathbb{R}^d$ where $d = 100$. The vector $w$ is also chosen uniformly from $S^{d-1}$ and $w'$ is an orthogonal vector to the hyperplane. High-dimensional sinusoid admits a linear classifier with accuracy $\approx 80\%$.

In the cases of datasets (i), (ii), and (iii) we created the train and tests sets by relabeling the train and test sets of MNIST and CIFAR10 with $\{0, 1\}$ labels according to the specific dataset (and excluded the images that are not relevant). All experiments are repeated 10 times with random initialization; standard deviations are reported in the figures (shaded area).

**Model details.**    Our results were consistent across various architectures and hyperparameter choices. For the Sinusoid distribution we train a 2-layer MLP, with ReLu activations. Each layer is 256 neurons. For the MNIST and CIFAR tasks in section 3 we train a CNN with 4 2D-Conv layers; each layer has 32 filters of size $3 \times 3$. After the first and second layers we have a $2 \times 2$ Max-Pooling layer, and at the end of the 4 convolutional layers we have two Dense layers of 2000 units. The activations on all intermediate layers are ReLUs. Across all architectures the last layer is a sigmoid neuron.

**Training procedure.**    We initialize the neural networks with Uniform Xavier [10]. We note that in all the experiments we use *vanilla SGD* without regularization (e.g. dropout) since we want to isolate and investigate purely the effect of the optimization algorithm. We use batch size of 32 for MLP's and 64 for CNNs. For the MLP's in section 3, the learning rate is 0.01. For the CNN's in Section 3, the learning rate is 0.001. For Section 4, we train the resnet and all smaller CNN's using SGD with momentum 0.9, batch size 128 and learning rate 0.01.

**Finding the Conditional Models.** For Section 3, in order to find the linear classifier $\ell$ that best explains the initial phase of learning, we do the following. For tasks (i), (ii), and (iv), where there

|  | Linear Model $\mu(F_{T_0}; L)/I(F_{T_0}; Y)$ | Null Model $\mu(F_{T_0}; \widetilde{L})/I(F_{T_0}; Y)$ |
|---|---|---|
| MNIST | 0.79 | 0.52 |
| CIFAR (ii) | 0.80 | 0.31 |
| CIFAR (iii) | 0.74 | 0.02 |
| Sinusoid | 0.74 | 0.27 |

Table 2: Performance Correlation of Linear Model vs. Null Model

|  | Simple Model $\mu(F_{T_0}; G_i)/I(F_{T_0}; Y)$ | Null Model $\mu(F_{T_0}; \widetilde{G}_i)/I(F_{T_0}; Y)$ |
|---|---|---|
| 2-layer CNN | 0.68 | 0.20 |
| 4-layer CNN | 0.72 | 0.31 |
| 6-layer CNN | 0.72 | 0.40 |

Table 3: Performance Correlation of simple CNN Models vs. Null Model on CIFAR First vs. Last 5

exists a unique optimal linear classifier, we set $\ell$ by training a linear classifier via SGD with logistic loss on the training set of the distribution.

For task (iii), we need to break the symmetry among many linear classifier that are roughly of equal performance. Here, we set $\ell$ by training a linear classifier via SGD to reproduce the outputs of $f_{T_0}$ on the training set. That is, we label the train set using $f_{T_0}$ (outputting labels in $\{0, 1\}$), and train a linear classifier on these labels. Note that we could also have trained on the sigmoid output of $f_{T_0}$, not rounding its output to $\{0, 1\}$; this does not affect results in our experiments.

In Section 4, we perform a similar procedure: After training $f_\infty$, we train simple models $g_i$ on the predictions of $f_\infty$ on the train set, via SGD.

**Estimating the Mutual Information.** Let $f, g$ be two classifiers, and $y$ be the true labels. In order to estimate our mutual-information metrics, we use the empirical distribution of $(F, G, Y)$ on the test set. For example, to estimate $I(F; Y|G)$ we use the definition

$$I(F; Y|G) = \sum_{(f,y,g) \in \{0,1\}^3} p(f, y, g) \log \left( \frac{p(f, y|g)}{p(f|g)p(y|g)} \right) \tag{6}$$

where $p(f, y, g)$ is the joint probability density function of $(F, Y, G)$ and $p(f|g)$, etc. are the conditional density functions. To estimate this quantity, we first compute the empirical distribution of $(f, y, g) \in \{0, 1\}^3$ over the test set. Let $\hat{p}(f, y, g)$ be this empirical density function. Then we estimate $I(F; Y|G)$ by evaluating Equation 6 using $\hat{p}$ in place of $p$.

**Further Quantitative Results.** In Tables 2 and 3 we provide further quantitative results for our experiments in Sections 3 and 4 respectively.

 **C   Additional Plots**

(a) Step 0        (b) Step 10        (c) Step 100        (d) Step 1000        (e) Step 10000

Figure 6: SGD training on a 3-layer, width-100 dense neural network. The data distribution is an (essentially) 1-dimensional Gaussian labeled by a linear classifier with 10% label noise. After a few hundreds of SGD steps the decision boundary is very non-linear outside of the training set, indicating that the neural network does not actually learn a linear classifier but rather a classifier that highly agrees with a linear one. In other words, there exist a linear classifier that explains the predictions of the neural network well, despite the fact that the network itself is highly non-linear.

(a) Step 0        (b) Step 10        (c) Step 100        (d) Step 1000        (e) Step 10000

Figure 7: A simplified version of Figure 1. SGD training on a 3-layer, width-100 dense neural network. The data distribution is an isotropic Gaussian in 2-dimensions, labeled by a linear classifier with 10% label noise. The blue line corresponds to the decision boundary of the neural network which becomes more "linear" in the initial stages before starting to overfit to the label noise.

(a) Good Init (via 100 SGD steps)        (b) Bad Initialization        (c) Continuing training from bad initialization.

Figure 8: Two different initializations, each with 88% accuracy on the training set. The data distribution is an isotropic Gaussian in 2-dimensions, labeled by a linear classifier with 10% label noise. The good initialization is found after running 100 SGD steps from a random initialization. The bad initialization is found by generating randomly labeled points, and fitting the function to them together with the original training set. One can see that continuing training from the bad initialization allows us to overfit to the training set but does not improve the population accuracy at all. The model here is again a 3-layer, width-100 dense neural network.