[Reviews · NeurIPS 2019]

Reviewer 1



This paper empirically shows that SGD learns functions of increasing complexity through experiments on real and synthetic datasets. Specifically, in the initial phase, the function learned can be almost explained by a linear function; in the later phase, the accuracy increases but the correlation between the learned function and the linear function does not decrease (so the linear function is not "forgot"). The measure used in quantifying these correlations is based on mutual information. This observation is extended beyond linear functions, to the case of CNNs with increasing depths, although the phenomenon is less clear there. Finally, the paper provides a theoretical result for a linear model, where overfitting doesn't hurt generalization. Originality: Measuring the function complexity using mutual information appears to be new in the study of deep learning. Quality: The experimental result is solid and the phenomenon is evident from the plots. The theoretical result appears to be correct, although I did not check the proof in appendix. Clarity: The paper is well written and easy to follow. Significance: Understanding the implicit regularization in training neural networks is an important problem. A key step is how to formulate the notion of implicit regularization, i.e., how to identify regularization as something concrete instead of "promoting generalization". This paper uses mutual information which is able to capture the interesting property of SGD identified in the paper. I think this observation is interesting and could be useful for further research in this topic. The downside is that the paper is purely empirical (the theoretical result in Sec 5 is only a toy example and not very relevant to other parts of the paper). There isn't any intuitive explanation of why the observed phenomenon might have happened. ==== update ==== I have read authors' feedback, and am going to keep my evaluation.

Reviewer 2



Update: I have read the author's response and am keeping my score =========== Originality: The idea that SGD biases networks towards simple classifiers has been discussed in the community extensively, but lacked a crisp formalization. This paper proposes a formal description for this conjecture (and also extends it -- Conjecture 1 and Claim 2) using a rich and simple information-theoretic framework. This work is roughly related to recent work on understanding the inductive bias of SGD in function space, e.g. Valle-Perez et al, Savarese et al: both only analyze the solution returned by SGD (under different assumptions), and not the intermediate iterates like this paper does. By defining the 'simplicity' of a neural network solely on the mutual information between its predictions and another classifier leads to objects which are invariant to reparameterization of the network, and only depend on the function that it implements. Such measure is highly novel and might prove to be a good candidate for the study of the complexity of networks in function space. The characterization of the two regimes (linear and 'post-linear') is also novel and interesting. -------------------- Quality: The claims are clear and backed up by experimental results. Since the paper aims at supporting the conjecture/claims with experiments, these could be more extensive. For example, plots of \mu_Y(F_t, L), I(F_t, L), I(F_t, Y_S) over time (as in Fig 4) in settings where the conjectured behavior does not manifest itself: - Bad initialization, such as in Figure 8b in the appendix - Different optimizer, which can be simulated with SGD + L2 regularization with small negative penalty lambda, as by enforcing that the weights grow any potential implicit L2 regularization of SGD is cancelled - Non-standard network parameterization In other words, it would be interesting to have some idea on what exactly is necessary for the conjectured behavior to happen. -------------------- Clarity: The paper is very well written, with clear definitions, experiments and discussion. -------------------- Significance: The formalization of the conjecture, the clear experimental results to back it up, and the proposed complexity measure \mu_Y(F, L) are all strong contributions, and very timely when considering the recent discussion on the inductive bias of SGD in the function space of neural nets. The proposed framework is very suited for this discussion as it is completely agnostic to how the network is parameterized. [1] Guillermo Valle-PĂ©rez, Chico Q. Camargo, Ard A. Louis - Deep learning generalizes because the parameter-function map is biased towards simple functions [2] Pedro Savarese, Itay Evron, Daniel Soudry, Nathan Srebro - How do infinite width bounded norm networks look in function space?

Reviewer 3



The main claim of the paper is that SGD learns, when training a deep network, a function fully explainable initially by a linear classifier. This, and other observations, are based on a metric that captures how similar are predictions of two models. The paper on the whole is very clear and well written. Importantly, the topic of the implicit bias induced by SGD and other optimizers is of great interest and significance to the community. However, the first two claims of the paper are not novel based on my understanding of the field. I also have some theoretical issues with the developed metric. More detailed comments: 1. Novelty of the first two claims of the paper are not clear to me. The third claim seems novel and intruiging. 1a) The first (main) claim does not seem novel. "Understanding training and generalization in deep learning by fourier analysis" and the cited "On the spectral bias of neural networks" try to theoretically and empirically show that SGD training a deep network learns sequentially from the lowest frequencies to the highest frequencies in the input space. More precisely, if one decomposes function of the network using Fourier transformation, then coefficients of the lowest frequency component will be learned first. This to me seems to undermine novelty of the paper because if the claim of these papers is correct, then it implies that initially a network is similar to a linear model. 1b) The claim that network learns function of increasing complexity is not very much formalized. Considering the level to which it is formalized it does not seem novel. See for instance https://arxiv.org/pdf/1610.01644.pdf. This makes outcome of the experiment not surprising. 2. I have two issues with the metric used. 2a) Some novelty is attributed to the proposed metric. But are there any benefit of using the proposed metric compared to a straightforward mutual information between network predictions (F) and linear classifier (L), i.e. I(F, L)? Perhaps it is trivial (sorry then if I missed it), but in any case it should be discussed. It seems to me that if I(F, L) = 0, then the proposed metric is also 0. 2b) While I, roughly speaking, agree with the main claim, I also wonder if the metric actually proves the main claim that initial performance is fully explainable away by a linear model. Consider a hypothetical scenario. Let's say the model initially learns to classify correctly all the examples that *are not* correctly classified by the linear model. Isn't the metric invariant to such a situation, i.e. wouldnt \mu curve look similar? If it is, then we cannot claim based solely on it that model initially learns the same things as a linear model. Other comments: 3. There is a group of highly related papers on the analysis of deep linear models. Therein it is relatively easy to show that network learns from simplest correlations between input and output to the highest (see for instance "Exact solutions to the nonlinear dynamics of learning in deep linear neural networks"), and later work has shown theoretically that this implies that SGD learns increasingly complex functions https://arxiv.org/abs/1904.13262. I think it would be nice to cite these papers. Another group of related papers that might be worth citing shows that DCNNs are highly sensitive to perturbations in the fourier domain of the image, e.g. https://arxiv.org/pdf/1809.04098.pdf. This implies that there is a direction in the input space that models reacts linearly to. 4. I would like to mention again that I found very intriguing the claim that the network is sensitive to things learned in the beginning of training. It might be worth emphasizing in the revised version. There is also a related work I am aware of that has discussed this phenomenon: https://openreview.net/forum?id=HJepJh0qKX. 5. I am not sure the claims are sufficiently formal to put them as claims. This might be personal taste, but I feel that paper clarity would benefit from a cleaner separation of formal and informal arguments. Update Thank you for the well written rebuttal. I am very much convinced by your explanations of the metric properties, as well as your contextualization of the linear probe paper. Indeed the metric is better than mutual information. It might be useful to include this discussion in the paper? I would like also to clarify that I was wrong in the review to say that Fourier based analysis of training of deep networks (such as "On the spectral (...)" paper) explains, strictly speaking, your empirical observations. Having said that, I am still concerned regarding novelty of the empirical observations. Novelty of empirical observations is key, because the theoretical contribution of the submission is arguably limited (as also mentioned by another reviewer). To add evidence that empirical observations are of limited novelty, see Figure 1 from https://arxiv.org/pdf/1807.01251.pdf that shows that low frequencies on MNIST and CIFAR-10 are learned first. This implies that a model close to linear (in some strict sense) exists in the beginning of training. Based on this I would like to keep my score.

[Author Response · NeurIPS 2019]

We thank all the reviewers for their insightful comments and suggestions. We will add citations and discussions of all the suggested related works in the full version. Reviewer-specific comments follow.

**Reviewer 1** We consider our mutual information framework to be a core contribution of our paper. In particular, our formalization of "how well a linear classifier explains the performance of a model" has many advantages over prior formalizations (e.g. see our response to Reviewer 3). Regarding our theory example (Section 5): We separate the question of why SGD learns simple concepts first (Claim 1) from the question of why it does not forget them (Claim 2). Our current theory is only relevant to the second question, and it shows that in a simplified setting: SGD does not "forget" the simple component even when trained to completion, provided it somehow learns the simple component first. We argue that this simple example captures many properties of real settings (overparametrization, existence of non-generalizable ERMs) and hence is valuable as a step towards more general theory. Regarding Claim 1, it is true that we currently have no theoretical understanding of why linear learning occurs. We consider this one of the most important open questions of our paper, and we are attempting to make progress on this in ongoing work.

**Reviewer 2** Thank you for pointing out the relevant papers. We also agree that it is scientifically valuable to describe settings where these phenomenon fail to occur (bad initialization, bad architecture/parameterization, bad optimizer, or pathological distributions). We plan on including a separate section with such examples in the final version.

**Reviewer 3** Regarding novelty of our claims: Although the idea that SGD learns functions of increasing complexity has been informally floating in the community, our formalization has many advantages over prior formalizations, as described below. Notably, our metrics respect the data distribution, are independent of network-parameterization, are tractable to estimate in high dimensions, and are experimentally demonstrated for real-world distributions.

Concretely, regarding "On the spectral bias of neural networks" [1]: They consider measuring "simplicity" via the Fourier spectrum of the learnt functions. However, the Fourier decomposition is taken with respect to a *uniform* distribution on inputs. This notion is not as meaningful – if the data is not uniform (for example, natural images are certainly not uniform in pixel-space), then a function which is highly correlated with a linear function when restricted to the data distribution may appear highly non-linear with respect to the uniform distribution. And vice-versa – a function which is nearly linear *under the uniform distribution* may in fact be highly non-linear when restricted to the data distribution. Our metrics do not suffer from this issue – they are taken with respect to the true data distribution.

The synthetic experiments in [1] are all for 1-dimensional inputs, since to quote [1]: "explicitly evaluating the Fourier coefficients ... becomes prohibitively expensive for larger d (e.g. on MNIST)". Instead, their MNIST experiments are only heuristically related to the metrics they propose. In contrast, the metrics in our paper are tractable even for high-dimensional inputs, and we estimate them to high-precision on real datasets (MNIST and CIFAR).

Regarding "Understanding ... deep learning by Fourier analysis" [2]: This work also performs Fourier analysis with respect to the uniform distribution on inputs, and so suffers from the same issues as [1]. Moreover [2] *requires* that the input distribution is itself uniform to carry through the analysis[1]. That is, the theorems of [2] do not hold for non-uniform input distributions, such as images. The experiments in [2] are not relevant to our claims, since they conflate the Fourier transform in the spatial domain (i.e. 2D Fourier transforming the input image, treated as a function $\mathbb{R}^2 \to \mathbb{R}$) with the Fourier transform in function space. (i.e. Fourier transforming the classification function $\mathbb{R}^d \to \mathbb{R}$). Finally, the work of [3] is largely unrelated to our work. The authors of [3] study how the internal layers of a network vary with the *depth* of the layer, while we study how the end-to-end classification function evolves as a function of SGD steps.

Reviewer 3 brings up two concerns with the performance correlation metric. First: why do we use $\mu_Y(F; L)$ instead of simply $I(F; L)$? While it is true that $\mu_Y(F; L) \leq I(F; L)$, $\mu_Y$ captures the degree to which the information learned by $F$ *about* $Y$ is explained by $L$ – whereas $I(F; L)$ only captures the correlation of $F$ and $L$, regardless of whether this correlation is useful for predicting $Y$ or not. For example, consider if $F(x) = L(x) \cdot Bernoulli(p)$. That is, $F$ is a linear classifier $L$ with noisy outputs. Here, $I(F; L) \ll 1$, due to the noise in $F$. However, $\mu_Y(F; L) = I(F; Y)$, and thus our metric recovers the fact that all the performance of $F$ in predicting $Y$ is explained by the linear $L$. Second, Reviewer 3 describes a scenario where $F$ first learns to classify the examples that are incorrectly classified by a linear model $L$, and notes that $\mu$ treats this the same as learning the correct portion of $L$. This isn't a problem, though, because the incorrect portion of $L$ is exactly the correct portion of the classifier $1 - L$, which is also linear. Contrast this with the following scenario: the samples come from a *mixture* of $L$ and an uncorrelated nonlinear model $N$, and $F$ learns $N$ first. This is a true example of $F$ not learning the linear part of the distribution, and accordingly $\mu_Y(F, L)$ will equal zero. We will include formal examples to build intuition for our metric in the final version.

[1] On the Spectral Bias of Neural Networks (2018). By Rahaman, Baratin, Arpit, Draxler, Lin, Bengio, Courville.
[2] Understanding training and generalization in deep learning by Fourier analysis (2018). By Zhi-Qin John Xu.
[3] Understanding intermediate layers using linear classifier probes (2016). By Guillaume Alain and Yoshua Bengio.

## Footnotes

[1]For Equation (5) in `https://arxiv.org/pdf/1808.04295v4.pdf`, applying Parseval's Theorem.


[Meta-Review · NeurIPS 2019]

There is a lot of support for the paper in the reviews. While much "folklore knowledge" exists around implicit regularization of SGD (e.g. towards approx. linear models), the paper does a very good job formalizing and answering relevant questions in a fruitful, yet simple, information theoretic framework. Some suggestions of improvement should be taken seriously, but all in all the paper makes a valuable contribution towards understanding the interplay of optimization and representational power (types of functions).